# Social Norms Support the Protein Transition: The Relevance of Social Norms to Explain Increased Acceptance of Alternative Protein Burgers over 5 Years

**DOI:** 10.3390/foods11213413

**Published:** 2022-10-28

**Authors:** Marleen C. Onwezen, Muriel C. D. Verain, Hans Dagevos

**Affiliations:** Wageningen Economic Research, Part of Wageningen University & Research, 2595 BM The Hague, The Netherlands

**Keywords:** social environment, injunctive norms, protein transition, consumer acceptance

## Abstract

Developing alternative protein products—based on protein sources other than regular meat—is a possible pathway to counter environmental and health burdens. However, alternative proteins are not always accepted by consumers, and more research is needed to support a shift to more alternative proteins. Prior studies have mainly focused on individual drivers and perceptions; although we expect that social norms—the perceptions of the opinions of relevant others—are highly relevant in accepting alternative proteins. Online surveys were conducted among 2461 respondents in 2015 and 2000 respondents in 2019 (cross-sectional datasets); a subsample (*n* = 500) responded to both surveys (longitudinal dataset). We add to the literature by (1) demonstrating the added explanatory value of social norms beyond a range of individual drivers; (2) showing that this finding holds over time, and (3) comparing the impact of social norms across different dietary consumer groups. Meat lovers and flexitarians are more prone to follow social norms whereas meat abstainers are more prone to follow their individual attitudes and values. This study highlights the relevance of investigations beyond personal variables such as personal norms and attitudes and underscores the relevance of considering the social aspects of accepting alternative proteins.

## 1. Introduction

People eat various food products daily, for example, yoghurt and fruit for breakfast, a sandwich with chorizo and cheese for lunch, and dinner, comprising a beef burger with a salad. These examples align with current Western consumption patterns and demonstrate that high amounts of animal-based proteins are consumed daily [1].

The current yearly meat consumption in the Netherlands is around 39 kg per person [2], which is around 104 g a day (19–79 years of age, [3]). This is about a third above the recommended intake by the national dietary guidelines. Moreover, although there has been an increase in the plant-based alternatives offered in supermarkets, and a lot of media attention for the protein transition, the consumption of meat has remained rather stable over the past 10 years [2]. The amount of self-declared flexitarians does seem to have increased over the past decade, from 13% in 2011 to 43% in 2019 [4]. Taken together, the consumption of meat in the Netherlands is above the recommended dietary guidelines, and although there are positive trends in terms of self-declared flexitarians and amount of available plant-based alternatives, the consumption of meat has remained stable over the past ten years.

High consumption levels of animal-based products generally, and the consumption of processed and red meat specifically, are associated with negative effects on planetary and human health, among their detrimental impacts on other sustainability issues [5]. Replacing regular meat with alternative proteins is a high-potential strategy to support the protein transition from animal- to plant-based proteins [6,7]. Examples of alternative proteins are pulses, insects, seaweed, and cultured meat (e.g., [8]). Although the consumption of alternative proteins is associated with health and environmental advantages, they are neither always nor automatically accepted by consumers.

Due to the urgency of a protein transition, it is important to understand what causes this low acceptance of alternative protein products. A potentially interesting determinant to examine is social norms. The literature indicates the relevance of social norms (i.e., perceptions of approval of relevant others) in a range of domains [9], although also specifically for food consumption [10] and more specifically for meat reduction [11]. Patel and Buckland ([12], p. 9) explicitly address the essential role social norms can play in enabling a shift to more sustainable food choices: “The harmful environmental and health impacts of current rates of meat consumption present an urgent need for norms to shift toward healthier and more sustainable reduced meat diets.” However, how social norms influence consumer behavior regarding alternative protein products is not fully clear yet. As most consumers do not consume alternative proteins regularly, social norms might reflect an unsupportive social environment for consuming these products. The current study aimed to explore whether perceived social acceptance of alternative proteins affects the consumer acceptance of alternative protein products and whether social norms decelerate or accelerate the protein transition. We specifically add to the literature in three ways.

First, we explore *the relevance of social norms beyond a set of individual drivers*. Previous studies have mainly focused on individual drivers, and the role of social norms in accepting alternative proteins is under-researched [13]. More specifically, we hypothesize that consumer perceptions of the approval of relevant others (i.e., social norms) are among the main drivers of the acceptance of alternative proteins because the social approval of behaviors can be highly influential [10].

Second, we explore *the predictive value of social norms over time*. It has rarely been tested as to whether consumers have become more receptive to alternative proteins over the years [14]. Subsequently, it remains unclear whether social norms regarding the consumption of alternative protein products have changed over the years and how this influences the acceptance. The current study will add to the literature by revealing whether social norms shift over the years and whether they accelerate or decelerate the protein transition.

Third, we investigate whether *the effectiveness of social norms varies across consumer groups varying in dietary consumption*: meat lovers, flexitarians, and meat abstainers. Social norms might be more impactful when they reflect the perceived approval of the most relevant peers [10], and these perceptions might vary across these consumer groups because they have other relevant peers and perceptions. We add to the literature by including this explorative part to assess whether social norms have the same impact across these different consumer groups. Until now, the impact of social norms on accepting alternative proteins has not been explored across consumer groups. Additionally, studies have often focused on a single alternative protein (e.g., insects; see Onwezen et al. [13] for an overview); we add to the literature by including multiple alternative proteins (i.e., pulses, seaweed, fish, cultured meat, and insects).

## 2. Theoretical Framework

Social norms refer to the individuals’ perceptions of social approval [15]. People often eat in social contexts, and previous studies have indicated that the food choices of others powerfully affect their consumption decisions [10]. Specifically, regarding food choices, social norms are the perceived standards for appropriate consumption with regard to the social group one relates to. This might refer to amounts of foods or specific food choices [10].

The social norm literature distinguishes two types of social norms. Others’ behavior in people’s social environment (i.e., descriptive norms) and their opinions regarding appropriate actions (i.e., injunctive norms) strongly influence the decisions and actions of people. We focused on injunctive norms, as alternative proteins reflect a wide range of products including those that are not yet widely consumed such as insects, or products that are not yet on the market such as cultured meat [13] Injunctive norms focus on perceptions of others’ approval [15], which can also be applied to novel or future products.

Previous studies have suggested that social norms are highly relevant drivers of novel food acceptance [16,17]. It is difficult for consumers to form opinions on novel and innovative foods, and individuals will use contextual cues such as social norms to form their opinions. As alternative proteins could be perceived as a novel trend contrasting with traditional meal preparations and perceptions, we propose that social norms are relevant to accepting alternative proteins. We explore the association between injunctive social norms (referred to as social norms) and accepting alternative protein sources. As noted previously, we add to the literature in three main ways. The theoretical rationales and propositions of each of these are discussed in detail below.

### 2.1. Explanatory Value of Social Norms beyond Individual Drivers in Accepting Alternative Proteins

Previous studies suggest that consumers are sensitive to the behavior and approval of relevant others regarding the consumption of alternative proteins. However, a recent literature review indicates that the number of studies including social norms is limited and that most studies have focused on individual drivers [13]. Moreover, Onwezen and colleagues suggest that although the number of studies is limited, the few existing studies show promising results. In the context of alternative proteins, several studies explored the role of social norms in combination with a range of personal factors and found that social norms were the most relevant driver [18,19,20]. We add to the literature by exploring the added value of social norms beyond a set of individual drivers. Based on the literature, we included a range of individual drivers of the acceptance of innovative products, although we do not claim this to be comprehensive.

An influential theory including social norms is the theory of planned behavior (TPB). According to the TPB, intentions are determined by attitudes, social norms, and perceived behavioral control (i.e., the perception that one can perform a particular behavior; [21]. The TPB has been applied in some studies concerning alternative proteins. Onwezen and colleagues [19] applied it in the context of accepting various alternative proteins. Moreover, some studies are available on specific alternative proteins such as insects [18,22,23] and fish [24,25]. The findings generally suggest that all drivers—perceived behavioral control, social norms, and attitudes—are relevant to explaining acceptance. Therefore, we also included perceived behavioral control and attitudes in our study, aside from social norms.

Some studies (e.g., [26,27,28]) have added personal norms to the TPB. Personal norms refer to feelings of moral obligation and form the core of the norm activation model [29]. Personal norms are often shown to be relevant in pro-environmental domains [28,30]. A TPB model including personal norms was also recently applied to accepting insects [31] and indicated that personal norms relate to intentions beyond the TPB variables, confirming the findings in related domains [26,32]. Therefore, we also included personal norms.

Finally, ambivalence can be an important driver of the acceptance of alternative protein sources, as consumers can have positive and negative associations [33,34] with alternative proteins, resulting in mixed feelings. For example, consumers might believe alternative proteins are less tasty and more expensive than regular meat while simultaneously believing that these alternatives might be more sustainable and healthier [13]. We anticipate ambivalence to be relevant to alternative proteins, as associations with alternative proteins are more complex than simply being positive or negative. Therefore, we also included anticipated ambivalence. Taken together, we formulated our first hypothesis (H1, see Figure 1) that *social norms are a more relevant driver for the acceptance of alternative protein products than individual drivers of attitudes, perceived behavioral control, personal norms, and ambivalence.*

### 2.2. Social Norms and Their Predictive Value Regarding Accepting Alternative Proteins over Time

We aimed to reveal whether social norms were associated with a shift in the protein transition over time. Recent studies indicate that particularly flexitarianism, and to a lesser extent, vegetarianism, have been slowly but gradually mainstreaming during the past years in various European countries [35,36] including the Netherlands [3,37,38]. This may result in a boost in the perceived social norms and their association with accepting alternative proteins. However, such developments do not alter the fact that the current food environment and prevailing food culture remain predominantly meat centered, although (excessive) meat consumption is neither uncontroversial nor inalterable. A European study recently indicated that meat consumption accounted for 30% of the total calorie consumption [39], and specifically in the Netherlands, most individuals still consume meat frequently [3]. A substantial reduction in meat consumption and eliminating meat from the diet remain the consumption patterns of minority groups. This implies that eating meat is normative and that established social norms concerning meat consumption may therefore negatively impact the acceptance of alternative proteins.

Although most studies on social norms reflect the relevance of highlighting that a substantial proportion of relevant peers approves or performs the specific behavior, some studies have highlighted the relevance of minority groups. Prior research has shown that dynamic norms, that is, changing norms over time, can positively influence sustainable behavior [40]. Moreover, previous studies in related domains have demonstrated that social norms could be highly influential, even in the context of minority behaviors such as those of environmental activists and vegans [41]. With modeling techniques, a reflection of how innovations become norms reveals that if a minority can maintain its majority status within its own group, the social norms of minorities can also affect consumer acceptance [42]. This suggests that social norms indicating that meat-eating is “normal” can simultaneously exist with the social norms of minorities reflecting the “normality” of meat reduction, and these might be positively associated with accepting alternative proteins.

In a more explorative research question (RQ1), we aimed to explore whether injunctive social norms help explain the acceptance of alternative protein products over time. Thus, we can state whether social norms support or decelerate the protein transition.

### 2.3. Social Norms across Consumer Groups Varying in Dietary Consumption of Meat

Previous studies have revealed that consumer groups may vary considerably in their motives and drivers for food choices (e.g., [43,44]). A recent Dutch study indicated that consumer groups with different attitudes and personal norms regarding meat consumption also differed in their motives, meat consumption, and intentions to reduce meat consumption [3]. One method of meaningfully segmenting consumers is using dietary consumption, for example, showing differences in the motivations and demographics [45] and relating these to differences in environmental impact in terms of CO_2_ emissions between meat-consuming and flexitarian, vegetarian, and vegan diets [46]. The current study adds an explorative part in which we researched the relevance of social norms across consumer groups differing in dietary consumption. Based on the aforementioned studies, we propose that consumer groups vary in their meat consumption and relevant peers. As social norms might be more impactful when they reflect the perceived approval of the most relevant peers [10], we propose that social norms and their impact might also vary across consumer groups. More specifically, meat abstainers might, for example, have more friends and colleagues open to alternative proteins, resulting in different perceived social norms compared to meat lovers. Until now, the impact of social norms on accepting alternative proteins has not been explored across consumer groups. We aim to explore, using a second research question (RQ2), whether social norms have the same impact across consumer groups varying in the dietary consumption of meat, namely meat lovers, flexitarians, and meat abstainers.

## 3. Method

### 3.1. Research Design

Respondents were recruited via a research agency (MSI-ACI) and asked to complete an online survey. Note that this study is part of a larger study financed by the Ministry of Agriculture, Nature and Food Quality in the Netherlands. The measurement therefore also included a range of other measurements. This project also resulted in a second paper [14]. The papers had a different focus, and used different drivers and measures at a different levels (category level versus product level). In this paper, the respondents were randomly assigned to one of five conditions: fish, seaweed, pulses, insects, or cultured meat. These alternative proteins were selected based on the two dimensions described in prior studies [14,47]: plant- versus animal-based proteins and conventional versus novel sources of proteins. Accordingly, the proteins cover a wide range of conventional and novel proteins from plant- and animal-based sources.

The online survey was first conducted in 2015 and repeated in 2019. We asked respondents to reflect on a burger made from fish, pulses, insects, seaweed, or cultured meat, depending on the condition to which they were allocated. We selected a burger because this could be made from the various protein sources in our study. Thus, the product was comparable for the different conditions, decreasing the probability that the respondents reflected on the products with different associations, products, and occasions in mind, for example, sushi for seaweed; pulses, beans, and tomato sauce for breakfast; and insects as a fried snack. Considering a burger as a joint product also aligns with a recent body of consumer studies including burgers of different origins [48,49,50].

### 3.2. Cross-Sectional and Longitudinal Data

Note that we had two types of datasets as a subsample of the respondents (*n* = 500; approximately 100 for each of the five conditions), who answered the online questionnaire in 2015 and 2019. We separated these samples to ensure that the respondents were either in the cross-sectional or the longitudinal dataset. The *cross-sectional datasets* included different respondents in 2015 and 2019 (without the subjects who answered the survey in both years), and the *longitudinal dataset* included the respondents who answered the survey in both years. For each analysis, we specify which dataset we used.

### 3.3. Participants

The Dutch cross-sectional samples comprised 2461 respondents in 2015 (male: 58.9%) with a mean age of 46.0 (*SD =* 15.8), and 2000 respondents in 2019 (male: 50.6%) with a mean age of 43.08 (*SD* = 15.8). The longitudinal dataset included the respondents who answered the survey in both years (47.4% male; mean age of 56.93 [*SD* = 12.6]).

### 3.4. Measurements

All items were measured on 7-point Likert scales, and all scales have been shown to perform well based on factor and reliability scores. The Cronbach’s alphas and Spearman–Brown correlations (for two items) are listed in Table 1.

*Social norms* were measured using three Likert-scale items. The items asked whether the respondents believed their family, friends, and/or colleagues wanted them to buy the specific burger (ranging from 1 “completely disagree” to 7 “completely agree”).

The variables of the TPB were measured following the work of Ajzen [21]. The *attitude* toward buying specific products was measured using three semantic differential scales with the following labelled endpoints: bad/good, negative/positive, and unfavorable/favorable. *Perceived behavioral control* was assessed through three items. The respondents indicated whether they felt they could buy a burger; whether it was their decision to buy a burger; or whether, if they wanted to, they could buy a burger on a 7-point Likert scale (ranging from 1 “completely disagree” to 7 “completely agree”).

*Personal norms.* Items were selected from previous research using the NAM [28]. The items were “I feel that I should …” and “Because of my own values/principles, I feel an obligation to behave in a {healthy or environmentally friendly} way”, with answer options ranging from 1 “completely disagree” to 7 “completely agree”.

*Ambivalence* was measured using the following three evaluative semantic differential scales based on a study by Priester and Petty [33]: “If I bought/used this product, I would experience no conflicting feelings/I would experience maximum conflicting feelings; I would feel no uneasiness at all/I would feel maximum uneasiness; I have no mixed feelings/I have strong mixed feelings”.

*Intention* was measured using two items (“I intend to buy the burger” and “I believe it is likely that I will buy the burger”) on a 7-point Likert scale (ranging from 1 “completely disagree” to 7 “completely agree”).

*Self-reported dietary meat consumption* was measured using the following item: “are you a vegetarian, vegan, or flexitarian?” The answering categories were as follows: no, I do not consume meat every day consciously/unconsciously, I am a flexitarian, I am vegetarian, I am vegan, and “other”. This variable was used to segment consumers into self-reported dietary groups: consumers who answered “no” were identified as meat lovers; those who did not consume meat every day or self-identified as flexitarian as flexitarians; and the vegan and vegetarian consumers as meat abstainers.

## 4. Results

### 4.1. Sample Description: Perceptions and Intentions over Time

Before starting our in-depth tests of the hypothesis and research questions, we performed descriptive analyses to obtain an impression of the trends over time for each specific driver and intention. For the cross-sectional datasets, separate *t*-tests for each alternative protein source were performed to assess changes in the included drivers and intention between the 2 years (Table 2). The results indicate that consumers generally perceived low social norms regarding all alternative proteins (below scale averages). Moreover, the means of intention were shown to be below the scale average for all alternative proteins. Generally, attitudes, perceived behavioral control, personal norms, and ambivalence showed higher scores (compared to intentions and social norms), indicating that consumers had relatively positive attitudes, personal norms, perceptions of control, and mixed feelings regarding these alternative proteins, whereas the perception of injunctive norms and intentions was generally lower. Differences also existed across alternative proteins; for example, insects were evaluated less positively on almost all drivers and intention than the other proteins.

Social norms, several individual drivers, and intentions were included as dependent variables, and the year was the independent variable. For all alternative protein sources, the social norm increased over time (see Table 2). Far fewer differences were observed for the other selected drivers of intention across the years. For pulses and cultured meat, all drivers showed a significant increase between the years, except for environmental personal norms for pulses and both personal norms for cultured meat. Perceived behavioral control increased for insects over the years. For intentions, the results showed that the respondents were more willing to eat all the burgers from different protein sources in 2019 compared to 2015. Note that we also performed the analyses for the longitudinal dataset. The results revealed much fewer differences in the mean scores within individuals due to a lower sample size. The results revealed no significant differences for the drivers of fish, seaweed, pulses, and insects. The results, however, did reveal a significant difference for perceived norms (*T*(1, 199) = 6.201 *p* < 0.05) and intention to buy cultured meat (*T*(1, 199) = 4.811 *p* < 0.05).

### 4.2. Explanatory Value of Social Norms beyond a Set of Individual Drivers on Intention to Consume Alternative Protein Burgers (Cross-Sectional Data)

We then aimed to test our first hypothesis (H1). We aimed to reveal whether social norms were indeed a more relevant driver of intentions to buy alternative proteins beyond a range of individual drivers. Separate hierarchical regression analyses were performed for each alternative protein source for the 2019 cross-sectional dataset. Intentions were included as a dependent variable and all drivers as independent variables.

The results (Table 3) confirm our hypothesis, revealing that social norms show the strongest association with the intention to buy the various protein burgers beyond a range of individual drivers. The TPB variables of attitudes and perceived behavioral control were significantly associated with the intention to buy burgers made from the various alternative proteins. Personal norms showed no significant association, and ambivalence was significantly associated with the intention to buy seaweed, insects, and cultured meat.

### 4.3. Relevant Drivers to Explain Variation in Intentions over the Years (2019–2015)

Second, we aimed to explore our more explorative research question (RQ1) regarding how social norms were associated with variation in intentions across years. Consistent with prior studies [51,52], we calculated the difference scores (2019–2015) to explore how social norms and the other individual drivers were associated with variation in acceptance over time. This method is useful for understanding how social norms are associated with variations in consumer acceptance over time. However, we do not state that this method is superior to other methods, as the literature describes a range of methods to explore differences across time (e.g., [53,54]). To ensure we drew the correct conclusions, we also explored the data with two other types of analyses (linear regression forecasting and a hierarchical regression with the year as dummy), as shown in the footnote and Appendix A (Table A1). Generally, these analyses showed comparable results.

We used the longitudinal dataset to calculate the difference scores for all variables, and regression analyses similar to those previously mentioned were performed on the difference scores. The results indicate that social norms were the most relevant driver for explaining changes in intentions between 2015 and 2019 (Table 4) for all alternative proteins. Social norms showed a positive association, indicating that more positive perceptions of social norms result in increased intentions over time.

Regarding the individual drivers, the results revealed that attitudes (toward fish, insects, and cultured meat) and perceived behavioral control (regarding seaweed, insects, and in vitro meat) were also shown to be relevant to understanding the variation in intentions to buy alternative protein burgers between the years. Ambivalence resulted in a significant increase in the explained variance of intentions to buy seaweed, and personal norms regarding the environment for the intentions to buy fish.

To ascertain that our finding reflect the data, we also performed two different types of analyses. We also performed the same hierarchical regression analyses with drivers measured in 2015 and intention in 2019 (linear regression forecasting). These results also revealed that social norms were the strongest explanatory variable for intentions to try a burger from fish (β = 0.509; *F*(6106) = 7.870 ***; R^2^ = 0.321), pulses (β = 0.482; *F*(6101) = 6.640 ***; R^2^ = 0.295), insects (β = 0.327; *F*(6, 81) = 3.628 **; R^2^ = 0.225), and cultured meat (β = 0.209; *F*(6, 99) = 5.527 ***; R^2^ = 0.263) in 2019. In this analysis, seaweed showed a different pattern, where the attitudes (β = 0.537; *F*(6106) = 6.566 ***; R^2^ = 0.283) and personal health norms (β = 0.243) showed a significant association with intention. Finally, we also performed a hierarchical regression analyses on a pooled dataset, with the year included as a dummy variable, all drivers as independent variables, and intentions as the dependent variable. In this manner, we could explore the impact of the drivers beyond the impact of year (or variation across time). The results showed for all alternative proteins that the drivers were more relevant in explaining the intentions to try a burger compared to the year. Moreover, the results also showed that social norm was the strongest explanatory variable of intentions for all alternative proteins (see Appendix A for all the details).

### 4.4. Variation across Groups: Meat Lovers, Flexitarians, Meat Abstainers

We also performed the cross-sectional analyses separately for consumer groups to explore whether social norms had the same impact for meat lovers, flexitarians, and meat abstainers (see measurements).

First, we explored whether significant differences existed in the perceived social norms between these three groups. The groups represented different group sizes, violating the assumption of the homogeneity of variance. Therefore Brown–Forsythe and Welch F-tests were conducted, with Games–Howell post hoc tests. Social norm was included as the dependent variable and the dietary consumer group as the independent variable. The results revealed that these consumer groups perceived different social norms, *F*(21987) = 8.424; *p* < 0.001. Post hoc analyses revealed that meat lovers showed a significantly lower perceived social norm (*M* = 2.28, *SD* = 1.55) than the flexitarians (*M* = 2.59, *SD* = 1.67). The meat abstainers showed no significant differences from the other groups (*M* = 2.53, *SD* = 1.85).

We then explored whether social norms had the same impact on accepting alternative protein burgers across groups. The regression analyses mentioned in the previous sections (range of drivers as the independent variables and intention as the dependent variable) were performed separately for each consumer group (Table 5). The results revealed that social norms, attitudes, and perceived behavioral control were significantly associated with the intentions for each consumer group. In accordance with our hypotheses, the results also revealed that social norms had a strong impact for each consumer group. For meat lovers and flexitarians, again, social norms had the strongest association with intentions. For meat abstainers, personal drivers of attitudes, perceived behavioral control, and personal health norms were also relevant. This indicates that vegetarians, who already have decided to avoid meat, are less sensitive to social norms and are more prone to following their individual drivers.

## 5. General Discussion

Changes in food consumption patterns are urgently needed to reduce the environmental and health burdens of high levels of animal-based protein consumption. Developing novel products based on alternative proteins is one option to address this need. However, these novel products are not always accepted by consumers. The literature suggests that the approval of such alternative protein products by relevant others (injunctive social norms) might be an important factor explaining this acceptance and that these norms might play an essential role in supporting the protein transition [12]. In the current paper, we examined how social norms influenced the consumer intentions to buy various novel alternative protein products (burgers made from fish, pulses, seaweed, insects, and cultured meat) over time and across consumer groups. We thus gained insights into the role of social norms compared to individual drivers in influencing consumers to shift from meat choices to alternative protein options (i.e., the protein transition). Before concluding this paper in Section 5.5 and addressing some limitations of the present work in Section 5.4, the three main findings are discussed in Section 5.1, Section 5.2 and Section 5.3.

### 5.1. The Strength of Social Norms

The most prominent finding is the strength of injunctive social norms (i.e., the perception that peers view the consumption of alternative protein burgers as the right thing to do) regarding one’s intention to buy alternative protein burgers. We found that social norms had the strongest association with the intention to consume these burgers compared to a range of individual drivers. This holds true for burgers from all of the included alternative protein sources (cross-sectionally) and over time.

Interestingly, an overview of the application of the TPB to food choices [55] revealed that, in most cases, attitudes were the strongest predictor of intentions, illustrating that accepting alternative proteins is a specific case. Social norms are possibly more relevant regarding alternative proteins because alternative proteins are perceived as a social behavior reflecting moral considerations for preserving the planet. Han and Stoel [56] stated that social norms are equally important as attitudes in the specific context of socially responsible behavior, whereas attitudes are often shown to be most relevant in other contexts. Another reason might be that consumers use more social information to form their opinions on novel foods or dietary patterns. This is consistent with prior research demonstrating that children use social learning to determine which foods are appropriate and safe to eat [10]. In the specific context of alternative proteins, consumers may use social norm information to form an opinion on these relatively unfamiliar foods or dishes.

The current findings indicate the relevance of including social norms in future research designs to understand and increase the consumer acceptance of novel alternative protein products. The acceptance of alternative proteins is not only a conscious deliberative process [14], but also an unconscious social process. The social environment is shown to greatly influence the consumers’ receptiveness to alternative proteins. Therefore, we recommend future research to further explore how social norms influence consumer acceptance by also including descriptive norms and social identity, as previous studies have demonstrated that norms are especially relevant when they reflect the norms of a group with which one identifies (e.g., [57]). Alternatively, focusing on dynamic norms that reflect changing social opinions might be especially relevant in the context of minority behaviors [40]. Another avenue for future research could be to investigate whether social norms can be used to increase the acceptance of alternative proteins and under which conditions these norms are most effective (e.g., norms related to an in-group or social identity, or positive norms rather than a negative norm reflecting that most people do not consume a particular product). Practitioners could use this information by providing indirect social information, for example, via enlarged shelf spaces or empty shelves [58] or by providing direct norm information on relevant peer groups (e.g., an increasing number of shoppers are buying this product [59]).

This research relates to a strand of literature emphasizing the importance of including other types of drivers besides individual-level rational drivers of food consumption such as interpersonal drivers [60], the food environment or eating context [59,61], and emotional drivers [28]. For example, Onwezen and colleagues [14] showed that positive emotions were the most important drivers of accepting alternative proteins over time. The current model can be used as a basis for future research so that a stepwise approach can show (a) which variables add to this model and (b) which relevant variations can be observed across alternative proteins (e.g., plant-based versus animal-based proteins), countries, consumer groups, and years.

### 5.2. Social Norms Accelerate the Protein Transition over Time

Our study is among the first to explore the acceptance of alternative proteins over time, revealing that social norms support the protein transition. In the literature, social norms are often mentioned as an influential mechanism, although only when most individuals perform a specific behavior [62]. Although this study shows that social norms are important in accepting alternative protein products, the absolute scores reflect that current social norms toward alternative protein products are low. The findings are consistent with a recent study by Verain and colleagues [3], who found low social norms toward meat reduction. The current study adds to the literature by revealing that although the social norm to consume alternative protein burgers is perceived as low, this norm still positively impacts accepting alternative proteins. This is consistent with recent research lines focusing on dynamic norms. These studies indicate that highlighting the trends of minority groups can support environmentally friendly behaviors [63]. Future research might further explore how low social norms or social norms of minority groups can influence behavior [64].

The results also suggest that consumers become increasingly open to alternative proteins over time, although this is not yet reflected in behavior. This finding is consistent with the literature, as consumers do not change their behavior overnight. They first become more receptive toward the specific behavior, and behavior slowly follows (e.g., from pre-contemplation to action [65]). More longitudinal studies are required to further determine how the protein transition develops over time and how social norms can be used to accelerate this transition. Interventions might help support the conversion from intentions to behavior change [66,67]. A possible way to use the potential of social norms is by communicating different implicit normative information such as defaults [68]. An example is making the “alternative” protein products the “regular” products, for example, in catering facilities, banqueting options, and on menus in the public (hospitals, universities, canteens at [local] governments) and private (restaurants, cafes) domains.

Despite the positive effects over time, the findings suggest that a shift toward more alternative proteins will be a long-term process given that social norms are important drivers of the transition to more sustainable diets [69] and an enabling social environment is vital for meat reduction to occur [11,35,70,71].

### 5.3. Social Bubbles: Variations in Explanatory Value of Social Norms across Dietary Groups

The results of the current paper suggest that social norms have different impacts on consumer acceptance across groups that differ in their meat consumption. The results indicate that individuals who consume meat are especially sensitive to social norms on alternative protein products. Meat abstainers who had already formed their opinions and refrained from eating meat showed the greater relevance of personal drivers such as attitudes and personal norms and appeared to be less sensitive to social norms. This could be explained by social norms on alternative proteins being more novel for meat lovers and flexitarians compared to vegetarians and vegans who have already internalized these alternative proteins in their diets. This pattern was also visible in nudging studies, showing that interventions are more effective for meat consumers than meat reducers [72].

Our results are promising in showing that even low social norms will potentially support the protein transition for meat lovers and flexitarians. This adds to the literature on social norms, as it implies that minorities will potentially accelerate sustainable transitions by inspiring other groups of individuals [41]. Future research might further disentangle these processes and explore the role of social norms for different consumer groups. For example, future research might explore how social norms operate among social bubbles of individuals with similar (versus distinct) consumption patterns.

### 5.4. Limitations Resulting in Directions for Future Research

The current paper had limitations, reflecting opportunities for future research. First, social norms were measured based on injunctive social norms. Future studies might include multiple social norms such as descriptive and dynamic norms to further explore the role of the social environment within the context of alternative proteins or norms reflecting other meat-related dietary behaviors such as meat-avoiding norms or stigmas. Another aspect that might be relevant is the context in which social norms are perceived; the prominence and level of norms may vary across contexts. For example, meat consumption may be more approved in the context of a dinner, barbecue, or social event.

Second, real behavioral choices were not included in the current work. Future research could therefore develop methods to include real-time behavior choices, for example, focusing on the trade-offs individuals might make in the supermarket (meat versus a meat alternative).

Third, personal norms were measured at a different level of abstraction compared to the other variables. This might explain why they showed no significant association with intention. We chose a more abstract level because personal norms relate to morality or personal values. Future research could measure the other variables at a more general level to compare variables at the same level of abstraction (e.g., [73,74]).

Fourth, an explorative part was included to investigate variations across consumer groups. The included groups are representative of the Dutch population, although a more balanced sample would be preferable regarding comparisons. Particularly, the group of meat abstainers was relatively small. Future studies might recruit balanced groups of various consumers, resulting in a research design to further explore the differences in distinct dietary consumer groups and the role of norms.

### 5.5. Conclusions

Our study is among the first to examine the relevance of social norms in the context of the consumer acceptance of alternative proteins. We demonstrated that in comparison to a wide range of individual drivers, social norms proved to be the most relevant driver in accepting a range of alternative protein burgers that included fish, seaweed, pulses, insects, and in vitro meat. Moreover, we explored the role of social norms over time, revealing that a low social norm could accelerate transitions over time. Finally, we observed that the impact of social norms varied across consumer groups, representing different dietary behaviors. It was shown that meat lovers and flexitarians are especially sensitive to social norm information, whereas vegetarians and vegans are more sensitive to individual drivers.

## Figures and Tables

**Figure 1 foods-11-03413-f001:**
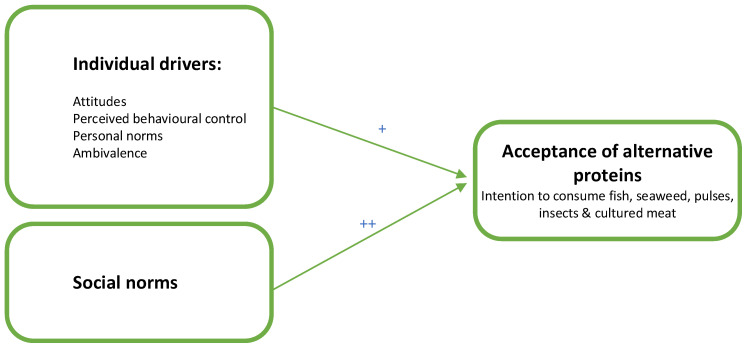
Conceptual model of the proposed associations. ***Note that we propose a stronger association (++) for social norms with acceptance compared to individual drivers (+)***.

**Table 1 foods-11-03413-t001:** Cronbach’s alphas for constructs of both samples.

	Cross-Sectional Datasets		Longitudinal Dataset	
	2015	2019	2015	2019
	α	α	α	α
Social norms	0.96	0.96	0.97	0.98
Attitude	0.96	0.96	0.97	0.97
Perceived behavioral control	0.73	0.74	0.73	0.76
Personal norm health	0.90	0.85	0.91	0.89
Personal norm environment	0.92	0.91	0.93	0.90
Ambivalence	0.96	0.95	0.96	0.96
Intention to buy	0.93	0.95	0.95	0.96

**Table 2 foods-11-03413-t002:** Differences in the mean scores across alternative proteins and years for intentions and its drivers.

	Fish		Seaweed		Pulses		Insects		Cultured Meat	
	2015	2019	2015	2019	2015	2019	2015	2019	2015	2019
Social norms	**1.90**	2.29 ***	2.06	2.30 *	1.98	2.45 ***	1.69	1.96 **	2.43	2.99 **
Attitude	4.15	4.33	4.54	4.61	4.23	4.63 ***	3.55	3.52	4.19	4.48 *
Perceived behavioral control	4.70	4.70	4.04	4.17	4.08	4.41 **	3.50	3.71 *	4.16	4.43 *
Personal norm health	5.04	5.00	4.99	4.96	5.03	5.22 *	4.99	4.98	5.06	5.07
Personal norm environment	4.86	4.95	4.87	4.98	4.94	5.07	4.89	5.03	4.93	5.02
Ambivalence	4.70	4.69	4.43	4.42	4.31	4.67 **	2.95	3.03	3.84	4.10 *
Intention	1.99	2.31 *	2.30	2.65 **	2.29	2.69 **	1.69	2.07 **	3.00	3.41 **

Note. * *p* < 0.05; ** *p* < 0.01; *** *p* < 0.001.

**Table 3 foods-11-03413-t003:** Hierarchical regression analyses for 2019 with drivers as the independent variables and intention as the dependent variable.

	Fish		Seaweed		Pulses		Insects		Cultured Meat	
	β	*t*	β	*t*	β	*t*	β	*t*	β	*t*
Social norm	**0.62 *****	**19.47**	**0.60 *****	**19.13**	**0.58 *****	**17.83**	**0.58 *****	**16.69**	**0.39 *****	**11.59**
Attitude	0.19 ***	4.68	0.11 *	2.43	0.25 ***	5.05	0.09 *	1.97	0.16 **	3.04
PBC	0.09 **	2.84	0.15 ***	4.39	0.06	1.85	0.12 **	3.17	0.31 **	7.52
PN health	−0.04	−1.09	0.06	1.89	0.04	1.02	0.01	0.36	0.02	0.56
PN environment	0.05	1.57	0.00	−0.06	−0.04	−0.99	−0.02	−0.45	−0.05	−1.35
Ambivalence	0.07	1.75	0.09 *	1.99	0.02	0.50	0.10 *	2.17	0.09 *	2.08
*F* = (df1, df2); R^2^	*F*(6, 499) = 107.914 ***; R^2^ = 0.566 ***		*F*(6, 500) = 134.525 ***; R^2^ = 0.620 ***		*F*(6, 499) = 105.014 ***; R^2^ = 0.556 ***		*F*(6, 498) = 103.647 ***; R^2^ = 0.558 ***		*F*(6, 499) = 123.843 ***; R^2^ = 0.602 ***	

Note. * *p* < 0.05; ** *p* < 0.01; *** *p* < 0.001; PBC = perceived behavioral control; PN = personal norm. The bold results refer to the highest associations between the range of drivers and intention.

**Table 4 foods-11-03413-t004:** Hierarchical regression analyses for the difference scores (2019–2015) with the drivers as independent variables and intention as the dependent variable.

	Fish		Seaweed		Pulses		Insects		In Vitro	
	β	*t*	β	*t*	β	*t*	β	*t*	β	*t*
Social norm	**0.33 *****	**4.08**	**0.57 *****	**7.23**	**0.28 ****	**3.01**	0.17	1.60	**0.45 *****	**5.58**
Attitude	0.30 **	3.01	−0.14	−1.43	0.01	0.06	0.22 *	2.07	0.32 **	3.43
PBC	0.13	1.64	0.17 *	2.23	0.17	1.68	0.24 *	2.31	0.20 *	2.43
PN health	0.02	0.26	−0.02	−0.17	−0.12	−1.09	0.17	1.54	−0.10	−1.12
PN environment	0.18 *	2.19	−0.08	−0.93	0.09	0.88	0.12	1.03	−0.09	−0.98
Ambivalence	0.09	0.88	0.25 *	2.62	0.22	1.81	0.09	0.87	0.01	0.06
*F* = (df1, df2); R^2^	*F*(6, 106) = 10.574 ***; R^2^ = 0.388 ***		*F*(6, 106) = 17.154 ***; R^2^ = 0.507 ***		*F*(6, 101) = 5.164 ***; R^2^ = 0.246 ***		*F*(6, 81) = 6.602 ***; R^2^ = 0.346 ***		*F*(6, 99) = 20.002 ***; R^2^ = 0.567 ***	

Note. * *p* < 0.05; ** *p* < 0.01; *** *p* < 0.001; PBC = perceived behavioral control; PN = personal norm. The bold results refer to the highest associations between the range of drivers and intention.

**Table 5 foods-11-03413-t005:** Hierarchical regression analyses for 2019 with drivers as independent variables and intention as the dependent variable for meat lovers, flexitarians, and meat abstainers.

	Meat Lovers (*n* = 1230)		Flexitarians (*n* = 685)		Meat Abstainers (*n* = 73)	
	β	*t*	β	*t*	β	*t*
Social norm	**0.60**	**29.43**	**0.60 *****	**21.61**	0.21 *	2.27
Attitude	0.18 ***	6.22	0.17 ***	4.35	**0.34 ***	2.57
PBC	0.11 ***	4.98	0.14 ***	4.68	**0.32 ****	3.26
PN health	0.00	0.04	0.02	0.75	0.16	1.68
PN environment	0.00	0.22	−0.03	−1.02	0.02	0.24
Ambivalence	0.03	1.19	0.02	0.59	0.02	0.17
Model 1 *F* = (df1, df2); R^2^	*F*(6, 1223) = 278.613 ***; R^2^ = 578 ***		*F*(6, 678) = 156.068 ***; R^2^ = 0.576 ***		*F*(6, 66) = 18.786 ***; R^2^ = 0.631 ***	

Note. * *p* < 0.05; ** *p* < 0.01; *** *p* < 0.001; PBC = perceived behavioral control; PN = personal norm. The bold results refer to the highest associations between the range of drivers and intention.

## Data Availability

Data is contained within the article.

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
