# Peer review of "Social Norms Support the Protein Transition: The Relevance of Social Norms to Explain Increased Acceptance of Alternative Protein Burgers over 5 Years"

_foods, 2022, doi:10.3390/foods11213413_

Round 1

Reviewer 1 Report

Thank you for the opportunity to review this paper. I have two comments that I hope can help the authors improve the paper.

1. It would be better to provide a figure that shows a graphical representation of your model.

2. I wonder why the authors did not pool the data collected in two years and simultaneously analyze them by including a year dummy and the moderating variables. Such an estimation method could provide clearer results regarding the effect of time and individual characteristics without having to conduct a series of t-tests. 

Author Response

We would like to thank the reviewer for carefully reading our manuscript and providing us with valuable suggestions to improve the manuscript. A detailed response can be found in the attachment. 

Reviewer 2 Report

This paper explores the impact of social norms on the acceptance of alternative proteins. The paper offers interesting insights in some respects, but requires some substantial improvements.      The main concern I have is the lack of econometric model and background.

Furthermore, some detailed comments are as follows:

1.      As the title included “explain increased acceptance of alternative protein burgers over 5 years”, the change of protein consumption in the Netherlands over five years should be included in background. I think a revised paper should lay out the context better

2.      A table should be provided to show the characteristics of samples. For example, the paper show that social norm boost the protein transition, but readers do not know what is sample’s social norm?

3.      There is unlabeled sign “”.

4.      Econometric model should be provided, because it is hard to understand the meaning of results.

5.      Why are fish, pulses, seaweed, insects and cultured meat as alternative protein products? The background information should be listed to help understand consumers’ dietary structure.

6.      Why do not provided more information about the variation of social norms, personal value and attitude over time.

Author Response

(The authors gave the same response as above.)

Round 2

Reviewer 2 Report

There are no further comments.